# Interval Hypoxic Training Enhances Athletic Performance and Does Not Adversely Affect Immune Function in Middle- and Long-Distance Runners

**DOI:** 10.3390/ijerph17061934

**Published:** 2020-03-16

**Authors:** Won-Sang Jung, Sung-Woo Kim, Hun-Young Park

**Affiliations:** 1Physical Activity and Performance Institute, Konkuk University, 120 Neungdong-ro, Gwangjin-gu, Seoul 05029, Korea; jws1197@konkuk.ac.kr (W.-S.J.); kswrha@konkuk.ac.kr (S.-W.K.); 2Department of Sports Medicine and Science, Graduate School, Konkuk University, 120 Neungdong-ro, Gwangjin-gu, Seoul 05029, Korea

**Keywords:** interval hypoxic training, hemodynamic function, autonomic nervous system balance, exercise performance, immune function, competitive middle- and long-distance runners

## Abstract

This study evaluated the effects of intermittent interval training in hypoxic conditions for six weeks compared with normoxic conditions, on hemodynamic function, autonomic nervous system (ANS) function, immune function, and athletic performance in middle- and long-distance runners. Twenty athletes were divided into normoxic training (normoxic training group (NTG); *n* = 10; residing and training at sea level) and hypoxic training (hypoxic training group (HTG); *n* = 10; residing at sea level but training in 526-mmHg hypobaric hypoxia) groups. All dependent variables were measured before, and after, training. The training frequency was 90 min, 3 d per week for six weeks. Body composition showed no significant difference between the two groups. However, the HTG showed more significantly improved athletic performance (e.g., maximal oxygen uptake). The hemodynamic function (e.g., oxygen uptake, oxygen pulse, and cardiac output) during submaximal exercise and ANS function (e.g., standard deviation and root mean square of successive differences, high frequency, and low/high frequency) improved more in the HTG. Immune function parameters were stable within the normal range before and after training in both groups. Therefore, hypoxic training was more effective in enhancing athletic performance, and improving hemodynamic and ANS function; further, it did not adversely affect immune function in competitive runners.

## 1. Introduction

Endurance exercise performance is related to various factors that can be altered by diverse hypoxic training methods, including erythropoiesis, exercise economy, capillary density, hemodynamic function, and acid-base response in the skeletal muscle [1,2]. Enhancing these factors, which are related to endurance exercise performance, increases the efficiency of aerobic energy production and consequently enhances maximum oxygen uptake (VO_2_max). It also enhances athletic performance by improving time until fatigue and increasing exercise intensity [3,4,5,6]. In particular, endurance exercise performance is reported to be the most affected by hemodynamic function, which is an indicator of oxygen transport and utilization ability [7].

Currently, altitude/hypoxic training is a common and popular practice for enhancing athletic performance in normoxic conditions among various athletes [8]. The most typical altitude/hypoxic training regimens proposed include living high-training high (LHTH), living high-training low (LHTL), and living low-training high (LLTH) methods. The LHTH method involves living and training at 1500–4000 m in natural altitude environments, while the LHTL method involves living at or near sea level but training under a natural or simulated altitude condition of 2000–3000 m [3,4,6]. The LLTH method may be of particular interest to athletes because this training commonly involves shorter hypoxic exposure (approximately two to five sessions per week of <3 h), lower cost, lesser effort, and lesser time than the LHTH and LHTL methods [1]. Further, the LLTH method, includes interval hypoxic training (IHT), repeated sprint training in hypoxia, and resistance training in hypoxic conditions; it has become an increasingly popular altitude/hypoxic practice, where athletes live at or near sea level but train in 2000–4500 m simulated hypobaric or normobaric hypoxic conditions [9,10,11].

Among the various LLTH methods, IHT consists of repeated exposures to 5–7 min of steady or progressive hypoxia, interrupted by equal periods of recovery; it can modify oxygen transport and energy utilization and induce permanent modifications in cardiac function [12]. Short-term repeated exposure to hypoxic conditions with high-intensity exercise enhances athletic performance via the metabolic and oxygen utilizing capacity [11,13,14,15]. However, some studies have not supported the enhancing effect of high-intensity training in hypoxic conditions on athletic performance [16,17,18,19]. These conflicting results are attributed to the fact that the enhancement of athletic performance was not verified on the basis of changes in hemodynamic function.

Heart rate (HR) variability (HRV) is a widely used marker reflecting cardiac modulation by sympathetic and vagal components and autonomic nervous system (ANS) activity. Dynamic adjustments in cardiac and peripheral vascular control, including their regulation by the ANS, occur in response to rapid changes in the HR [20,21]. Change in HRV with exercise training have often been interpreted as increase in vagal activity or ANS balance function, which is related to athletic performance [21]. Herzig et al. [22] reported that HRV markers of vagal activity are moderately associated with athletic performance variables, such as 10-mi race time. Dong [23] explained that HRV was becoming one of the most useful tools for tracking the time course of exercise training adaptation of athletes and for setting the optimal exercise intensity that leads to enhanced athletic performance. Therefore, it is essential to examine the effectiveness of exercise training in hypoxic conditions with changes in HRV, which is useful in enhancing the athletic performance.

Exercise in hypoxic conditions acts as a stressor to yield greater physiological and metabolic functions than exercise in normoxic conditions, thereby causing changes in the neuroendocrine system and affecting immune function [24,25]. Further, exposure to hypoxic conditions stimulates the release of epinephrine in the adrenal medulla, increases the sympathetic nervous system activity, and increases the concentration of cortisol and adrenal cortical hormone in the blood [26,27]. The most representative changes in immune function following exposure to hypoxic conditions include decreased CD4+ T cell count; decreased T cell activation and proliferation; lymphocytosis; neutropenia; and inflammatory upregulation of cytokines, such as interleukin (IL)-6, IL-1, C-reactive protein, and tumor necrosis factor (TNF)-α [25,28,29,30,31]. As described above, exposure to hypoxic conditions results in a change in immune function based on various changes in the physiological, metabolic, and neuroendocrine systems. However, studies on changes in immune function following exercise training in hypoxic conditions are scarce.

Considering that various hypoxic training regimens are commonly used to enhance athletic performance in normoxic conditions based on hematological and non-hematological changes, it is important to examine the effects on immune function in terms of health and conditioning. Moreover, the World Anti-Doping Agency is concerned that various hypoxic training regimens can have a potentially negative effect on health [32]. Thus, an essential task for elite athletes is to examine how exercise training in hypoxic conditions affects their immune function, and establish the efficacy and stability of hypoxic training.

Therefore, this study aimed to investigate the effects of intermittent interval training on hemodynamic function, ANS function, immune function, and athletic performance of competitive middle- and long-distance runners in a hypoxic condition versus that in a normoxic condition. We hypothesized that intermittent interval training in a hypoxic condition would enhance hemodynamic function, ANS function, and athletic performance more than in a normoxic condition, and would not adversely affect immune function in competitive middle- and long-distance runners.

## 2. Materials and Methods

### 2.1. Subjects

Subjects, whose characteristics are presented in Table 1, were men and were competitive, moderately trained, middle- and long-distance runners (*n* = 20) registered with the Korea Association of Athletics Federations. They were assigned equally to the normoxic (NTG) and hypoxic (HTG) training group based on their body composition and athletic performance. We explained the experiment and possible adverse effects before the start of the study to participants and obtained their signed informed consent to participate in this study. This study was approved by the institutional review board of Konkuk University (7001355-2020002-HR-359) and was conducted in accordance with the provisions of the Declaration of Helsinki.

### 2.2. Study Design

The study design is shown in Figure 1. Twenty athletes were equally divided into the NTG (*n* = 10; intermittent interval training in a normoxic condition; 760 mmHg) and HTG (*n* = 10; intermittent interval training in a hypoxic condition; 526 mmHg; simulated altitude of 3000 m). All testing and training were performed in a 9-m (width) × 7-m (length) × 3-m (height) chamber with a temperature of 22 ± 1 °C and humidity of 50 ± 5% regulated by an environmental control chamber (NCTC-1, Nara control, Seoul, Korea).

The present study comprised a 5-day pre-test period (i.e., 3 testing days and 1 rest day between the testing days), 6-week training period under each environmental condition, and 5-day post-test period. The post-test period began 3 d after the final training session.

On the first pre- and post-testing days, blood samples were collected between 8:00 and 10:00 a.m. after 12 h of fasting for the analysis of blood variables related to immune function in the normoxic condition. Thereafter, body composition and ANS function were measured. Subsequently, the VO_2_max was measured to evaluate exercise performance in the afternoon. On the second pre- and post-testing days, hemodynamic function parameters were measured during a 30-min bout of submaximal cycle ergometer exercise. The exercise intensity was set at individual cycle ergometer exercise load values corresponding to 80% maximal HR (HRmax) obtained during the pre-test period. On the third testing day, a 3000-m time trial record was measured on an authorized track stadium at sea level.

All athletes performed the following in 90-min sessions: Warm-up, interval training, and cool-down. The training frequency was 90 min, 3 d per week for 6 weeks. Warm-up and cool-down were set at 50% HRmax for each participant for 5 min, which was then increased by 10% HRmax every 5 min and performed for 15 min. The interval training sessions consisted of 10 repetitions of interval running exercise (5 min of exercise corresponding to 90–95% HRmax and 1 min of rest) on a treadmill.

All exercise training sessions in the hypoxic conditions were supervised by directors, coaches, and the researchers.

### 2.3. Blood Composition

Body composition parameters, such as weight, free fat mass, and percentage body fat were analyzed using Inbody 770 (Inbody, Seoul, Korea).

### 2.4. Hemodynamic Function

Hemodynamic function was measured before and after training while the participants performed submaximal cycle ergometer exercise corresponding to 80% HRmax obtained during the pre-test period for 30 min at sea level [1,11]. The oxygen uptake (VO_2_) was measured using the K5 auto metabolism analyzer (Cosmed, Rome, Italy) and a breathing valve in the form of a facemask. The HR, stroke volume index (SVi), and cardiac output index (COi) were assessed non-invasively using a thoracic bioelectrical impedance device (PhysioFlow PF-05, Paris, France). The oxygen pulse (O_2_ pulse) was calculated as VO_2_/HR.

### 2.5. ANS Function

ANS function was assessed by measuring HRV. After approximately 10 min of rest, four pads were placed on the wrists and ankles using an HRV meter (LAXTHA; CANS-3000, Daejeon, Korea), and participants’ HRV was measured in the resting condition. The following parameters were evaluated: Standard deviation (SD) of successive differences (SDNN) and root mean square of successive differences (RMSSD) for the time domain methods and low frequency (LF) band, high frequency (HF) band, and LH/HF band ratio for the frequency domain methods [33].

### 2.6. Immune Function

To assess immune function, white blood cell (WBC), eosinophil, neutrophil, basophil, natural killer (NK) cell, B cell and T cell counts were measured before and after the intervention. Three milliliters of blood were collected between 8:00 and 10:00 a.m. after 12 h of fasting. All blood samples were placed in an anticoagulant heparin tube and centrifuged at 3500 rpm for 10 min, and the serum was collected and rapidly frozen at −70 °C. Thereafter, the frozen or refrigerated serum was commissioned by the Clinical Laboratory of Green Cross Medical Foundation and analyzed using the method described below.

In detail, WBC, neutrophil, eosinophil, and basophil counts were measured via flow cytometry using a cellpack kit (Sysmex, Kobe, Japan). NK cell, B cell, and T cell counts were analyzed using FC500 (Beckman Counter, CA, USA) and measured via flow cytometry using an NK cell kit (Beckman Coulter, Paris, France), a CD19-PE kit (Beckman Coulter, Paris, France), and a CD3-PC5 kit (Beckman Coulter, Paris, France), respectively.

### 2.7. Athletic Performance

To evaluate athletic performance, VO_2_max was measured before and after the intervention with the modified BRUCE protocol for graded exercise testing on a treadmill (S25TX, SFET, Seoul, Korea) using a K5 breath by breath auto metabolism analyzer (K5, Cosmed, Rome, Italy). The graded exercise test was completed when the following three criteria were satisfied: (1) VO_2_ plateau: No further increase in oxygen use per minute even with an increase in work performed, (2) HR within 10 beats of the age-predicted HRmax: This is the basis for using participants’ HRmax as a surrogate for the VO_2_max when designing personal training programs, and (3) plasma (blood) lactate concentrations of >7 mmol/L.

The 3000-m time trial records were measured on a 400-m track at sea level in Seoul between 9:00 and 10:00 a.m. (temperature = 22–24 °C; humidity = 60–80%; wind = 0–10 km/h). To avoid the effect of racing strategies, all starts were staggered by at least 2 min.

### 2.8. Statistical Analysis

Means and SDs were calculated for each primary dependent variable. Normality of distribution of all outcome variables was verified using the Sharpiro-Wilk test. The two-way analysis (time × group) of variance with repeated measures of the “time” factor was used to analyze the effects of the training methods on each dependent variable. Partial eta-squared (*η2*) values were calculated as measures of the effect size. When a significant interaction effect was found, the Bonferroni post-hoc test was used to identify within-group changes over time. Additionally, the paired *t*-test was used to compare between the post- and pre-training values of the dependent variables in each group separately. A priori power analysis was performed with G-power for the energy metabolic parameter (VO_2_ during 30-min of submaximal exercise) based on previous research [1], indicating that a sample size of 14 participants (7 subjects per group) would be required to provide 80% power at an α-level of 0.05. We anticipated a dropout rate of >10% and aimed for a starting population of 20. All analyses were performed using the Statistical Package for the Social Sciences version 24.0 (IBM Corp., Armonk, NY, USA). The level of significance was set at 0.05 (a priori).

## 3. Results

### 3.1. Body Composition

Data on the body composition in both groups before and after training are shown in Table 2. No significant interaction was observed in all body composition parameters, i.e., body composition did not affect the change in the other dependent variables.

### 3.2. Athletic Performance

Figure 2 depicts the pre- and post-training data on athletic performance in both groups. There was a significant interaction for the VO_2_max (*η*^2^ = 0.686, *p* < 0.001). The post-hoc analysis revealed significant enhancements in both groups (NTG: *p* < 0.01, HTG: *p* < 0.001), and the improvement in VO_2_max was greater in the HTG than in the NTG (NTG: 1.5%, HTG: 6.3%). No significant interaction was observed for the 3000-m time trial records.

### 3.3. Hemodynamic Function

As shown in Table 3, there was a significant interaction for the VO_2_ (*η*^2^ = 0.251, *p* < 0.05), O_2_ pulse (*η*^2^ = 0.588, *p* < 0.001), and COi (*η*^2^ = 0.575, *p* < 0.001). Compared with the NTG, the HTG showed a significant decrease in the VO_2_ (*p* < 0.001) and a significant increase in the COi (*p* < 0.001) during submaximal exercise for 30 min. The O_2_ pulse (NTG: *p* < 0.01, HTG: *p* < 0.01) during submaximal exercise for 30 min significantly increased in both groups, and the improvement in O_2_ pulse was greater in the HTG than in the NTG (NTG: 17.7%, HTG: 24.7%). No significant interaction was observed for the HR and SVi.

### 3.4. ANS Function

Table 4 depicts the pre- and post-training data on HRV in both groups. Significant interaction was seen for all HRV parameters, including SDNN (*η*^2^ = 0.732, *p* < 0.001), RMSSD (*η*^2^ = 0.777, *p* < 0.001), LF band (*η*^2^ = 0.616, *p* < 0.001), HF band (*η*^2^ = 0.693, *p* < 0.001), and LF/HF band ratio (*η*^2^ = 0.420, *p* < 0.01). In the post-hoc analysis, the NTG showed a significant decrease in the average of all R wave to R wave intervals (mean RR; *p* < 0.001), SDNN (*p* < 0.01), RMSSD (*p* < 0.001), LF band (*p* < 0.01), and HF band (*p* < 0.001). In contrast, the HTG presented a significant increase in the mean RR (*p* < 0.01), SDNN (*p* < 0.01), RMSSD (*p* < 0.01), total power (*p* < 0.05), HF band (*p* < 0.01), and LF/HF band ratio (*p* < 0.01).

### 3.5. Immune Function

As shown in Table 5, there was a significant interaction for the WBC (*η*^2^ = 0.293, *p* < 0.05), neutrophil (*η*^2^ = 0.416, *p* < 0.01), monocyte (*η*^2^ = 0.580, *p* < 0.001), and B cell (*η*^2^ = 0.258, *p* < 0.05) counts. Compared with the NTG, the HTG showed a significant increase in the WBC (*p* < 0.05) and neutrophil (*p* < 0.01) counts and a significant decrease in the monocyte count (*p* < 0.001). Conversely, the B cell count significantly decreased (*p* < 0.05) in the NTG compared to that in the HTG. No significant interaction was observed for the eosinophil, basophil, NK cell, and T cell counts.

## 4. Discussion

In the present study, we hypothesized that intermittent interval training in a hypoxic condition (simulated 3000-m, 526-mmHg hypobaric hypoxia) versus that in a normoxic condition would enhance athletic performance and improve hemodynamic function and ANS function and would not adversely affect immune function in competitive middle- and long-distance runners. Our findings were consistent with these hypotheses.

### 4.1. Athletic Performance

Our study confirmed that intermittent interval training in a hypoxic condition improved VO_2_max more than that in a normoxic condition.

The oxygen transport capacity in systemic conditions is most often evaluated using VO_2_max [8]. Exercise training in a hypoxic condition may increase exercise performance by inducing various biochemical and structural adaptive changes in the skeletal and cardiac muscles, which favor the oxidative process and can enhance non-hematological parameters, such as exercise economy, acid-base balance, and metabolic response during submaximal exercise, ultimately leading to improved oxygen delivery and utilization capacity [1,8,9,10,11,13]. Among the various LLTH methods, IHT consists of repeated exposures to 5–7 min of steady or progressive hypoxia, interrupted by equal periods of recovery; it can modify oxygen transport and energy utilization and induce permanent modifications in the cardiac function [12]. The short-term repeated exposure to hypoxic conditions, with high-intensity exercise, enhances athletic performance via the metabolic and oxygen utilizing capacity [11,13,14,15].

However, research findings on IHT as an effective hypoxic training method for enhancing athletic performance at sea level are inconclusive [8]. In various previous studies, the difference in the enhancement of athletic performance, via the IHT method, was attributed to the intensity of the exercise performed in the hypoxic conditions [8,9,34]. There are also some differences in the type of exercise. Park and Lim [8] evaluated the effects of six weeks of hypoxic training on exercise performance in moderately trained competitive swimmers and reported that a moderate intensity of continuous and interval training in a hypoxic condition for six weeks resulted in an unclear change in the aerobic and anaerobic performance compared to normoxic training. They also argued that the unclear improvement after hypoxic training was attributed to the relatively low exercise intensity. Conversely, Czuba et al. [14] evaluated the effects of a three-week continuous hypoxic training with a relatively high exercise intensity corresponding to 95% lactate threshold workload on athletic performance in well-trained cyclists and reported a significant increase in exercise performance (e.g., VO_2_max, VO_2_ at the lactate threshold, maximal work load, and work load at the lactate threshold). Further, Roels et al. [35] observed a significant increase in the VO_2_max after seven weeks of high-intensity IHT compared to that after normoxic training. Regarding the inconsistency of previous research results, McLean et al. [34] suggested that the greater athletic performance (e.g., VO2max and time trial records) with IHT was more likely achieved if exercise training in hypoxic conditions is performed with high-intensity interval workouts. Our study also showed that intermittent interval training in a hypoxic condition was effective in enhancing athletic performance by increasing the VO_2_max of competitive runners compared to training in a normoxic condition. We believe that these positive results were attributable to IHT performed with high-intensity interval workouts.

### 4.2. Hemodynamic Function

Our study verified that intermittent interval training in a hypoxic condition improved VO_2_, O_2_ pulse, and COi during submaximal exercise more than that in a normoxic condition.

Hemodynamic function represents the dynamics of blood flow in systemic conditions, and the hemodynamic system continuously monitors and adjusts to the conditions in the body and its environment. Further, athletic performance is highly related to hemodynamic function, which serves as an indicator of oxygen transport and utilization capacity [13].

As mentioned earlier, various previous studies have reported that IHT effectively enhanced athletic performance by improving the metabolic responses (e.g., blood lactate level, glycolytic enzyme and glucose transport, and acid-base balance regulation) and oxygen utilization capacity [1,11,14,15]. IHT may increase exercise performance by inducing various biochemical and structural adaptive changes in the skeletal and cardiac muscles that favor the oxidative process and can enhance non-hematological parameters, such as exercise economy, acid-base balance, and metabolic response during submaximal exercise, ultimately leading to improved oxygen delivery and utilization capacity [1,8,9,10,11,13]. However, there is a lack of studies demonstrating that IHT enhanced athletic performance based on changes in hemodynamic function, which indicates oxygen delivery and utilization capacity in systemic conditions. Therefore, we examined the enhancement effect of intermittent interval training in the hypoxic condition on athletic performance in relation to hemodynamic function and found that compared to training in a normoxic condition, intermittent interval training in a hypoxic condition effectively improved hemodynamic function by decreasing the VO_2_ and COi and increasing the O_2_ pulse during submaximal cycle ergometer exercise for 30 min among competitive runners.

In the present study, the significantly improved hemodynamic function (significant decrease in the VO_2_ and COi and significant increase in the O_2_ pulse) indicates enhanced exercise economy, which is defined as the amount of energy per unit distance [4,36,37,38]. Exercise economy and VO_2_max are widely known as determinant factors of athletic performance [13,36,37,38]. IHT has been shown to enhance ATP re-synthesis (per 1 mole O_2_) and decrease ATP levels at a given exercise intensity (e.g., speed and workload) [36,38,39]. Exercise training enhances athletic performance by increasing the efficiency of oxygen transport and utilization and increases energy availability, which consequently improves the invigoration of the parasympathetic nervous system via the activation of β-adrenergic receptors in the cardiac muscles and efficiently changes cardiac function by increasing venous return [13,39]. Thus, the reduced COi and increased VO_2_ and O_2_ pulse during submaximal exercise in our study indicate improvement in the function of the heart as a pump for delivering oxygen and oxygen utilization and delivery to the muscle tissue [13,39]. Thus, the IHT in our study enhanced the VO_2_max and exercise economy, which are two important factors of athletic performance; this enhancement was probably affected by improvements in hemodynamic function.

### 4.3. ANS Function

Our study proved that intermittent interval training in a hypoxic condition improved successive differences (SDNN), root mean square of successive differences (RMSSD), high frequency (HF) band, and LF/HF band ratio. 

HRV is a widely used marker reflecting cardiac modulation by sympathetic and vagal components and ANS activity; it is the most sensitive and reproducible marker among those obtained from tests for measuring changes in the ANS [20,40,41]. Further, HRV is caused by the interaction between the sympathetic and parasympathetic nervous system on the sinoatrial nodes. Therefore, the HRV test is generally used in the field of mental health examination and health science [20,40]. Clinical application of HRV is mainly associated with the prediction of sudden cardiac infarction and assessment of progression of cardiovascular and metabolic illness [23]. Recently, the sports science field has been using the HRV test for monitoring exercise training effects and recovery [22,23,42].

Decreases and increases in the vagal-derived indices of HRV have been shown to indicate negative, and positive adaptations, respectively, to exercise training [42]. In elite athletes, HRV changes are highly related to the efficiency of exercise training, and positive adaptations, such as increased cardiovascular fitness after exercise training, have been reported to be associated with HRV [23,36,42]. Moreover, some previous studies reported that athletic performance (e.g., VO_2_max) correlated with improvement in HRV [23,43]. Therefore, our study examined the effect of intermittent interval training on athletic performance along HRV parameters in a hypoxic condition versus that in a normoxic condition. We found that all HRV parameters (e.g., SDNN, RMSSD, LF band, HF band, and LF/HF band ratio) showed a significant interaction between the NTG and HTG; the SDNN, RMSSD, HF band, and LF/HF band ratio indicated greater improvements in the HTG than in the NTG.

Among the HRV parameters, SDNN is an indicator of comprehensive HRV, has a high correlation with the HF band, and mainly reflects parasympathetic nervous system activity [44]. The RMSSD is an estimate of the short-term components of HRV, and the larger the value, the more physiologically healthy and relaxed it is [44,45]. The HF band mainly reflects the activity of the vagal nerve supplying the heart and is representative of parasympathetic nervous system activity [46]. Conversely, the LF band correlates with stress and reflects the sympathetic nervous system activity [47]. As the LF band increases, the overall HRV decreases and heart instability increases. The LF/HF band ratio reflects the overall balance of the ANS. A higher LF/HF band ratio indicates that the sympathetic nervous system is relatively activated or the parasympathetic nervous system activity is suppressed [21]. In the present study, the HTG showed more significant improvements in most HRV parameters (e.g., SDNN, RMSSD, HF band, and LF/HF band ratio) than did the NTG; this positive HRV adaptation may have influenced the enhancement of exercise performance. Further, our study showed that IHT could improve the training and recovery quality, and yield more efficient exercise training effect by improving HRV.

### 4.4. Immune Function

Our study confirmed that intermittent interval training in a hypoxic condition showed significant changes in WBC, neutrophil, monocyte, and B cell count, however, all parameters of immune function were altered after the two-week IHT was clinically within the normal range.

As mentioned above, the IHT regimen in our study enhanced the athletic performance with improvement in hemodynamic function and HRV; it is critical to examine the impact of the regimen on immune function in terms of the health and conditioning of athletes. The World Anti-Doping Agency is concerned that various hypoxic training regimens can have a potentially negative effect on health [32]. Therefore, it is an imperative task for elite athletes to examine how exercise training in hypoxic conditions affects immune function and to establish the efficacy and stability of hypoxic training.

Exercise in hypoxic conditions acts as a stressor to yield greater physiological and metabolic function than that in normoxic conditions, causing changes in the neuroendocrine system and consequently affecting immune function [24,25]. Representative changes in the nervous and endocrine systems by acute exposure to hypoxic conditions correspond to stimulating the release of epinephrine in the adrenal medulla, increasing the sympathetic nervous system activity, and increasing the concentration of cortisol and adrenal cortical hormones in the blood [26,27]. Changes in the neuroendocrine system by exposure to hypoxic conditions induce changes in immune function, such as decreased T cell count; decreased T cell activation and proliferation; increased neutrophil count; and upregulation of inflammatory markers, including IL-6, IL-1, C-reactive protein, and TNF-α [28,48]. As described above, exposure to hypoxic conditions has been reported to result in a change in immune function based on changes in the physiological, metabolic, and neuroendocrine systems. However, studies on changes in immune function following exercise training in hypoxic conditions are scarce.

Although there are differences in the regimen of hypoxic training, Tiollier et al. [49] investigated the impact of an 18-day LHTL training camp on secretory immunoglobulin A (sIgA) levels in 11 (six female and five male) elite cross-country skiers. There was a downward trend in the sIgA levels, which reached significance in the LHTL group but not in the control group. Further, the salivary IgA levels were still lower at baseline than those post-operatively. They strongly suggested a cumulative negative effect of physical exercise and hypoxia on the sIgA levels during LHTL training. Brugniaux et al. [50] examined the effect of LHTL training performed for 13–18 d through leukocyte count evaluation in 41 athletes from 3 federations (cross-country skiers, *n* = 11; swimmers, *n* = 18; runners, *n* = 12) and found that the leukocyte count was not affected, except at 3500 m. Park et al. [51] recently reported a case study in which an LLTL regimen was used to evaluate the effects of a 2-week hypoxic training on immune function in Korean national cycling athletes with disabilities. They found that all immune function parameters were in the normal range even after two weeks of hypoxic training.

In the present study, the HTG showed a more significant increase in the WBC and neutrophil counts and a significant decrease in the monocyte count than did the NTG. Conversely, the B cell count significantly decreased in the NTG compared to that in the HTG. However, all immune function parameters that were altered after the two-week IHT were clinically within the normal range. Thus, the six-week IHT in this study did not negatively affect the immune function of the competitive runners, which is consistent with the results of a previous study [51].

## 5. Limitation of the Study

Some limitations of our study should be considered when interpreting our results. Although the present study was designed systematically with equally controlled experiments, small sample sizes were a limitation to check the effects of an intermittent interval training in a hypoxic condition versus that in a normoxic condition on athletic performance, hemodynamic function, ANS function, and immune function in middle- and long-distance competitive runners. Thus, larger samples are warranted in future studies to access sports field practice. Furthermore, the athlete’s dietary intake and conditioning were not investigated.

## 6. Conclusions

Our study confirmed that intermittent interval training in a hypoxic condition for six weeks would enhance the athletic performance and improve hemodynamic and ANS function; further, it did not adversely affect the immune function of competitive runners compared to that of runners training in a normoxic condition.

## Figures and Tables

**Figure 1 ijerph-17-01934-f001:**
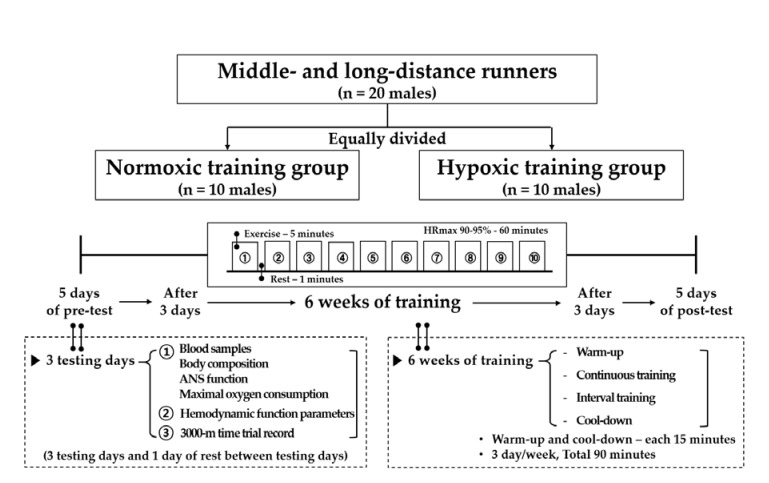
Study design of the present study.

**Figure 2 ijerph-17-01934-f002:**
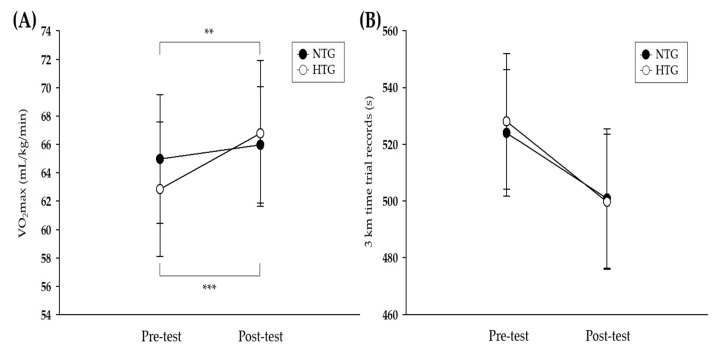
Athletic performance parameters before and after the 6-week exercise training. (**A**) Change in the VO_2_max after exercise training in each environmental condition. (**B**) Change in the 3-km time trial record after exercise training in each environmental condition. VO_2_max = maximum oxygen uptake, NTG = normoxic training group, HTG = hypoxic training group. * Significant difference between the pre- and post-tests, ** *p* < 0.01, *** *p* < 0.001.

**Table 1 ijerph-17-01934-t001:** Characteristics of the athletes.

Variables	NTG	HTG	*t*-Value	*p*-Value
Number (n)	*n* = 10	*n* = 10	-	-
Environmental condition (mmHg)	Sea level (760 mmHg)	3000-m simulated altitude (526 mmHg)	-	-
Age (year)	25.9 ± 1.2	26.3 ± 1.5	−0.499	0.624
Height (cm)	176.9 ± 7.6	178.2 ± 3.5	−0.514	0.616
Weight (kg)	70.8 ± 5.8	71.2 ± 6.3	−0.490	0.630
BMI (kg/m^2^)	23.1 ± 1.5	22.8 ± 0.9	−1.554	0.138
FFM (kg)	51.1 ± 4.4	52.1 ± 4.8	−0.490	0.630
Percent body fat (%)	17.5 ± 2.7	18.4 ± 1.8	−0.882	0.389

Values are expressed as means ± standard deviations. NTG = normoxic training group, HTG = hypoxic training group, BMI = body mass index, FFM = free fat mass.

**Table 2 ijerph-17-01934-t002:** Changes in the body composition of the competitive runners before and after the tests.

Variables	NTG (*n* = 10)		*p*-Value	HTG (*n* = 10)		*p*-Value	*p*-Value (*ɳ*^2^)		
	Pre	Post		Pre	Post		Group	Time	Interaction
Weight (kg)	70.8 ± 5.8	71.1 ± 5.8	0.146	71.2 ± 6.3	71.1 ± 7.3	0.945	0.555 (0.020)	0.233 (0.078)	0.269 (0.067)
BMI (kg/m^2^)	23.1 ± 1.5	23.1 ± 1.6	0.105	22.8 ± 0.9	22.6 ± 0.9	0.390	0.136 (0.119)	0.138 (0.118)	0.919 (0.001)
FFM (kg)	51.1 ± 4.4	50.6 ± 4.4	0.147	52.1 ± 4.8	52.1 ± 5.5	0.952	0.555 (0.020)	0.233 (0.078)	0.269 (0.067)
Percent body fat (%)	17.5 ± 2.7	17.3 ± 2.9	0.274	18.4 ± 1.8	17.8 ± 2.0	0.275	0.477 (0.028)	0.144 (0.115)	0.549 (0.020)

Values are expressed as means ± standard deviations. NTG = normoxic training group, HTG = hypoxic training group, BMI = body mass index, FFM = free fat mass.

**Table 3 ijerph-17-01934-t003:** Changes in hemodynamic function during submaximal cycle ergometer exercise for 30 min among the competitive runners before and after the tests.

Variables	NTG (*n* = 10)		*p*-Value	HTG (*n* = 10)		*p*-Value	*p*-Value (*ɳ*^2^)		
	Pre	Post		Pre	Post		Group	Time	Interaction
HR (beat/30 min)	5375.7 ± 431.4	4946.1 ± 287.8	0.003	5207.4 ± 308.7	4830.5 ± 267.6	0.001	0.296 (0.060)	0.000 (0.674) ^†††^	0.694 (0.009)
VO_2_ (mL/30 min)	1355.6 ± 114.4	1322.9 ± 118.9	0.094	1194.1 ± 139.5	1118.1 ± 141.8	<0.001 ***	0.005 (0.364) ^††^	0.000 (0.678) ^†††^	0.024 (0.251) ^†^
O_2_ pulse (mL/beat/30 min)	553.5 ± 111.2	651.5 ± 49.2	0.004 **	551.4 ± 129.1	687.7± 35.9	0.007 **	0.614 (0.014)	0.418 (0.037)	0.000 (0.588) ^†††^
SV_i_ (mL/beat/30 min)	1691.5 ± 99.4	1660.7 ± 102.5	0.018	1520.4 ± 117.4	1518.2 ± 188.1	0.971	0.007 (0.342) ^††^	0.579 (0.017)	0.629 (0.013)
CO_i_ (L/30 min)	244.7 ± 20.8	253.7 ± 15.4	0.280	281.7 ± 22.1	227.2 ± 31.4	<0.001 ***	0.526 (0.023)	0.002 (0.411) ^††^	0.000 (0.575) ^†††^

Values are expressed as means ± standard deviations. NTG = normoxic training group, HTG = hypoxic training group, HR = heart rate, VO_2_ = oxygen consumption, O_2_ pulse = oxygen pulse, SVi = stroke volume index, COi = cardiac output index. Significant interaction or main effect: ^†^
*p* < 0.05, ^††^
*p* < 0.01, ^†††^
*p* < 0.001; Significant difference between the pre- and post-tests: ** *p* < 0.01, *** *p* < 0.001.

**Table 4 ijerph-17-01934-t004:** Changes in ANS function among the competitive runners before and after the tests.

Variables	NTG (*n* = 10)		*p*-Value	HTG (*n* = 10)		*p*-Value	*p*-Value (*ɳ*^2^)		
	Pre	Post		Pre	Post		Group	Time	Interaction
SDNN (ms)	61.0 ± 4.5	50.1 ± 6.8	0.001 **	56.6 ± 14.4	66.9 ± 11.0	0.001 **	0.157 (0.108)	0.827 (0.003)	0.000 (0.732) ^†††^
RMSSD (ms)	37.6 ± 7.7	23.7 ± 4.9	<0.001 ***	32.5 ± 10.4	44.8 ± 12.9	0.002 **	0.055 (0.190)	0.646 (0.012)	0.000 (0.777) ^†††^
LF band (ms^2^)	7.1 ± 0.6	6.7 ± 0.4	0.001 **	7.0 ± 0.7	7.2 ± 0.6	0.052	0.399 (0.040)	0.013 (0.294) ^†^	0.000 (0.616) ^†††^
HF band (ms^2^)	6.5 ± 0.4	5.9 ± 0.3	<0.001 ***	6.0 ± 1.1	6.9 ± 1.0	0.003 **	0.455 (0.031)	0.353 (0.048)	0.000 (0.693) ^†††^
LF/HF band ratio	1.2 ± 0.1	1.3 ± 0.2	0.207	1.5 ± 0.5	1.2 ± 0.3	0.008 **	0.447 (0.032)	0.012 (0.304) ^†^	0.002 (0.420) ^††^

Values are expressed as mean ± standard deviation. ANS = autonomic nervous system, NTG = normoxic training group, HTG = hypoxic training group, SDNN = standard deviation of the NN interval, RMSSD = root mean square of successive differences, LF = low frequency, HF = high frequency. Significant interaction or main effect: ^†^
*p* < 0.05, ^††^
*p* < 0.01, ^†††^
*p* < 0.001; Significant difference between the pre- and post-tests: ** *p* < 0.01, *** *p* < 0.001.

**Table 5 ijerph-17-01934-t005:** Changes in immune function among the competitive runners before and after the tests.

Variables	NTG (*n* = 10)		*p*-Value	HTG (*n* = 10)		*p*-Value	*p*-Value (*ɳ*^2^)		
	Pre	Post		Pre	Post		Group	Time	Interaction
WBC count (10^3^/µL)	5.4 ± 0.5	5.6 ± 0.6	0.124	4.6 ± 0.8	6.4 ± 2.1	0.013 *	0.977 (0.000)	0.003 (0.387) ^††^	0.014 (0.293) ^†^
Eosinophil count (%)	3.8 ± 1.5	3.7 ± 1.6	0.623	3.4 ± 1.3	3.1 ± 1.4	0.153	0.459 (0.031)	0.126 (0.125)	0.286 (0.063)
Neutrophil count (%)	48.5 ± 5.8	46.4 ± 4.8	0.312	44.6 ± 4.6	53.0 ± 11.1	0.004 **	0.635 (0.013)	0.047 (0.201) ^†^	0.002 (0.416) ^††^
Basophil count (%)	1.3 ± 0.5	1.2 ± 0.6	0.434	0.8 ± 0.1	0.6 ± 0.1	<0.001	0.004 (0.385) ^††^	0.018 (0.273) ^†^	0.343 (0.050)
Monocyte count (%)	9.2 ± 2.6	9.8 ± 2.0	0.141	9.1 ± 0.7	7.3 ± 0.7	<0.001 ***	0.090 (0.151)	0.023 (0.255) ^†^	0.000 (0.580) ^†††^
NK cell count (%)	22.0 ± 4.9	21.3 ± 4.9	0.603	19.6 ± 5.1	19.0 ± 5.6	0.622	0.280 (0.064)	0.467 (0.030)	0.988 (0.000)
B cell count (%)	14.6 ± 2.9	13.1 ± 3.8	0.048 *	16.7 ± 2.0	17.6 ± 1.6	0.229	0.008 (0.331) ^††^	0.548 (0.020)	0.022 (0.258) ^†^
T cell count (%)	66.6 ± 8.0	65.9 ± 10.3	0.633	70.1 ± 6.9	71.1 ± 8.1	0.126	0.249 (0.073)	0.780 (0.004)	0.256 (0.071)

Values are expressed as means ± standard deviations. NTG = normoxic training group, HTG = hypoxic training group, WBC = white blood cell, NK= natural killer. Significant interaction or main effect: ^†^
*p* < 0.05, ^††^
*p* < 0.01, ^†††^
*p* < 0.001; Significant difference between the pre- and post-tests: * *p* < 0.05, ** *p* < 0.01, *** *p* < 0.001.

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
