# Peer review of "Interval Hypoxic Training Enhances Athletic Performance and Does Not Adversely Affect Immune Function in Middle- and Long-Distance Runners"

_ijerph, 2020, doi:10.3390/ijerph17061934_

Round 1
Reviewer 1 Report
The article is well written, provides thorough literature review, methods are well selected and the conclusions are supported by the results. The article also presents some interesting results. My only comment is that for clarity, I would also add the p-values to the Tables as interpretating when t or F values are significant or close to significant is challenging.
Author Response
Point-by-Point Responses to the Reviewers
We thank the reviewers for their guidance for further improving our revised manuscript (ijerph-731013) entitled “Interval hypoxic training enhances athletic performance and does not adversely affect immune function in middle- and long-distance runners”. As described below, we have responded to all the comments brought up by the reviewers and incorporated all the changes suggested by the reviewer. In addition, we have received English corrections from the English grammer editing company, Editage.
Reviewer 1.
Review Report Form
Comments and Suggestions for Authors
- The article is well written, provides thorough literature review, methods are well selected and the conclusions are supported by the results. The article also presents some interesting results. My only comment is that for clarity, I would also add the p-values to the Tables as interpretating when t or F values are significant or close to significant is challenging.
: Thanks for the good information and comments. As a reviewer 1 comments, we added p-values to all the tables. Thanks again for your good comment.

Reviewer 2 Report
While the topic of manuscript was interesting, it is unclear what the authors are trying to present and hard to read.
Abstract: Overall please show data with P-value for the results, rather than presenting a simplified sentence. For example, if it is significant; VO2max (number in pre +- SD to number in post +- SD; P value)
Line 16: Please mention what type of athletes were included
Line 20: Body composition showed no significant interaction between the groups at baseline? If so, please clarify if this is at baseline or post-training.
Line 23: ANS function: The examples are not clear what it is referring to. SD and square root of successive differences, high frequency etc. of what? It seems like heart rate variability is what it is refereeing to based on the rest of manuscript.
Line 25: what is immune function parameters indicating here? It is not consistent from prior sentences as other function was mentioned with examples. Please indicate what was measured for immune function.
Introduction
First paragraph: It is very unclear what the sentences are indicating, possibly due to grammatical issues and writing styles.
Line 42: probably better to focus on athletes here, not coaches and sports scientists as that’s what manuscript focuses on.
Line 43-44: Redundant as this is explained earlier
Line 45-46: Unclear what those LHTL LLTH are. Please describe briefly here.
Line 70-81: This may belong to discussion section.
Methods
Line 99: is ‘assigned’ meaning randomization based on body composition? Muscle mass or fat mass, BMI? Please clarify how randomization was performed. If not please justify.
Line 101-104: Please specify IRB approved number
Line 119: typically the hour of fasting is shown. A day meaning 24 hours fasting?
Line 137-144: Please reference the method.
Please specify power calculation and randomization. If this study was conducted as a pilot study, this has to be clarified throughout the manuscript.
Results
Table 2: the way how it is described is unclear, particularly F-value. Also why F-value instead of P-value? F-values for group and time shows significance at pre or post? Interaction means group x time interaction? Please revise. Probably significance column can be added right next to each group, rather than placing in the right corner.
3.2 Athletic performance: Please show the data in a table as well.
Discussion
Overall, please compare this study to other previous studies: what is different or consistent and why. It is still good to include the potential mechanisms though. As there are sub header under discussion, it would be better to summarize the main finding in the first paragraph of each sub header.
Need to present limitations of this study.
Line 265: How the intensity is different from this study? What was intensity parameter like Czuba et al study?
Line 358: Redundant, as shown previously.
Author Response
Point-by-Point Responses to the Reviewers
We thank the reviewers for their guidance for further improving our revised manuscript (ijerph-731013) entitled “Interval hypoxic training enhances athletic performance and does not adversely affect immune function in middle- and long-distance runners”. As described below, we have responded to all the comments brought up by the reviewers and incorporated all the changes suggested by the reviewer. In addition, we have received English corrections from the English grammer editing company, Editage.
Reviewer 2.
Comments and Suggestions for Authors
While the topic of manuscript was interesting, it is unclear what the authors are trying to present and hard to read.
: The present study aimed to investigate the effects of intermittent interval training in a hypoxic condition versus a normoxic condition on athletic performance with hemodynamic function, ANS function, and immune function in competitive middle- and long-distance runners. We demonstrated that interval hypoxic training vs interval normoxic training enhances athletic performance with improved hemodynamic function and ANS function, but does not adversely affect immune function. Given that WADA is concerned that various hypoxic training regimes can have a potentially negative effect on health, our findings are of great value for athletes in performing hypoxic training.
Abstract: Overall please show data with P-value for the results, rather than presenting a simplified sentence. For example, if it is significant; VO2max (number in pre +- SD to number in post +- SD; P value).
: Thanks for the good information and comments. However, IJERPH restricts abstracts to 200 words. Therefore, it is difficult to express statistical significance in the abstract section. Please understand these limitations.
Line 16: Please mention what type of athletes were included.
: We changed "competitive runners" to "middle- and long-distance runners".
Line 20: Body composition showed no significant interaction between the groups at baseline? If so, please clarify if this is at baseline or post-training.
: The body composition showed no significant interaction and no significant difference between training in each groups. See Table 2 for the details of the ANOVA analysis.
Line 23: ANS function: The examples are not clear what it is referring to. SD and square root of successive differences, high frequency etc. of what? It seems like heart rate variability is what it is refereeing to based on the rest of manuscript.
: Thanks for the good comments. Several previous studies have used heart rate variability as a biomarker for the ANS function. Therefore, our study used the HRV parameters as the ANS function. And the HRV parameters were measured at rest. This is described in detail in the method section.
Reference
Evans S, Seidman LC, Tsao JC, Lung KC, Zeltzer LK, Naliboff BD. Heart rate variability as a biomarker for autonomic nervous system response differences between children with chronic pain and healthy control children. J Pain Res. 2013 Jun 12;6:449-57. doi: 10.2147/JPR.S43849.
Hayano J, Yuda E. Pitfalls of assessment of autonomic function by heart rate variability. J Physiol Anthropol. 2019 Mar 13;38(1):3. doi: 10.1186/s40101-019-0193-2.
Elghozi JL, Girard A, Laude D. Effects of drugs on the autonomic control of short-term heart rate variability. Auton Neurosci. 2001 Jul 20;90(1-2):116-21.
Line 25: what is immune function parameters indicating here? It is not consistent from prior sentences as other function was mentioned with examples. Please indicate what was measured for immune function.
: To assess immune function, white blood cell (WBC), eosinophil, neutrophil, basophil, natural killer (NK) cell, B cell, and T cell counts were measured before and after the intervention. However, IJERPH restricts abstracts to 200 words. Therefore, we only mentioned the subvariables of the parameters that showed statistically significant differences.
Introduction
First paragraph: It is very unclear what the sentences are indicating, possibly due to grammatical issues and writing styles.
: Thanks for the good information and comments. We described the factors that affect endurance exercise performance and athletic performance in the first paragraph. Although we are not native English speakers, we think that we have wrote the necessary information appropriately. In addition, we have received English corrections from the English grammer editing company, Editage. If there is an appropriate paragraph example that reviewer 2 wants, we will match it.
Line 42: probably better to focus on athletes here, not coaches and sports scientists as that’s what manuscript focuses on.
: Thank you for your kind comment. We changed the sentence as follows: Currently, altitude/hypoxic training is a common and popular practice for enhancing athletic performance in normoxia among various athletes.
Line 43-44: Redundant as this is explained earlier
: As reviewer 2 comments, we have removed the following duplicate sentences: The endurance exercise performance of athletes in normoxia can be enhanced by various altitude/hypoxic training methods
Line 45-46: Unclear what those LHTL LLTH are. Please describe briefly here
: As reviewer 2 comments, we have added a description of LHTH, LHTL, and LLTH as follows: The LHTH method was the first design of living and training at 1500 - 4000 m in the natural altitude environments, and the LHTL is a method of living at or near sea level but training under a natural or simulated altitude condition of 2000 – 3000 m [3,4,6]. The LLTH method may be of particular interest to athletes because this training commonly involves shorter hypoxic exposure (approximately two to five sessions per week of < 3 hours), lower cost, less effort, and shorter time than the LHTH and LHTL methods [1]. And the LLTH method, including interval hypoxic training (IHT), repeated sprint training in hypoxia, and resistance training in hypoxia, has become an increasingly popular altitude/hypoxic practice, where athletes live at or near sea level but train at 2000-to-4500-m simulated hypobaric or normobaric hypoxic conditions [7-9].
Line 70-81: This may belong to discussion section.
: The sentence on “Line 70-81” is considered to be the basic explanation for the reason why the study examined the change of immune function according to interval hypoxic training vs interval normoxic training. Therefore, we believe that this sentence is necessary in the introduction section. However, if reviewer 2 insists that this sentence should be removed from the introduction section, we will remove it later.
Methods
Line 99: is ‘assigned’ meaning randomization based on body composition? Muscle mass or fat mass, BMI? Please clarify how randomization was performed. If not please justify.
: As shown in the manuscript, all athletes were equally assigned to a normoxic training group (NTG) or a hypoxic training group (HTG) according to their body composition (weight, BMI, FFM, and %body fat) and athletic performance (VO2max and 3km time trial records).
Line 101-104: Please specify IRB approved number
: This is our mistake. This study was approved by the institutional review board of Konkuk University (7001355-2020002-HR-359) and was conducted in accordance with the provisions of the Declaration of Helsinki.
Line 119: typically the hour of fasting is shown. A day meaning 24 hours fasting?
: Thanks for the good information and comments. In our study, fasting was performed for 12 hours. The sentence has been changed as follows: On the first pre- and post-testing days, blood samples were collected between 8:00 and 10:00 AM after 12 hours of fasting for analysis of blood variables related to immune function at the normoxic condition.
Line 137-144: Please reference the method.
: As reviewer 2 comments, we have added two references to how hemodynamic function is measured.
Please specify power calculation and randomization. If this study was conducted as a pilot study, this has to be clarified throughout the manuscript.
: As we explained in the text, means and SDs were calculated for each primary dependent variable. Normality of distribution of all outcome variables was verified using the Kolmogorov-Smirnov test. An a priori power analysis was performed with G-power for the energy metabolic parameter (VO2 during 30-min of submaximal exercise) based on previous research [1], indicating that a sample size of 14 participants (7 subjects per group) would be required to provide 80% power at an α-level of 0.05. We anticipated a more than 10% dropout rate and aimed for a starting population of 20.
Results
Table 2: the way how it is described is unclear, particularly F-value. Also why F-value instead of P-value? F-values for group and time shows significance at pre or post? Interaction means group x time interaction? Please revise. Probably significance column can be added right next to each group, rather than placing in the right corner.
: As reviewer 2 comments, we changed the F-value to the p-value. Also, we added the p-value for the difference between before- and after-training in each group. Naturally, Interaction means group x time interaction.
3.2 Athletic performance: Please show the data in a table as well.
: We have shown a change in athletic performance as a figure. Check out Figure 1.
Discussion
Overall, please compare this study to other previous studies: what is different or consistent and why. It is still good to include the potential mechanisms though. As there are sub header under discussion, it would be better to summarize the main finding in the first paragraph of each sub header.
: As reviewer 2 comments, we compare our study with other previous studies. We also discussed the potential mechanisms in the discussion section. In addition, we summarize the main results in the first paragraph of each subheader.
Need to present limitations of this study.
: We wrote the following limitations of the study: In our study, there are some limitations to consider when interpreting results. Although present studies have been designed systematically with equally controlled experiments, small sample sizes can be limited to check the effects of an intermittent interval training in a hypoxic condition vs a normoxic condition on athletic performance, hemodynamic function, ANS function, and immune function in middle- and long-distance competitive runners. More subjects may be needed in future studies to access sports field practice. Secondly, the athlete's dietary intake and conditioning were not investigated.
Line 265: How the intensity is different from this study? What was intensity parameter like Czuba et al study?
: This is our mistake. In Czuba et al. study, well-trained cyclists performed exercise training at an intensity equivalent to 95% lactate threshold workload. It is not 95% maximal heart rate. Compared with Czuba et al. study, our study shows higher exercise intensity.
Line 358: Redundant, as shown previously.
: This sentence has been used redundantly to emphasize the importance of assessing immune function in the application of IHT to athletes. We therefore believe that this sentence is necessary for this paragraph. Please understand.
Reviewer 3 Report
Dear authors,
In this article, you gave me the impression of trying to reinvent the wheel. It's a pity because at the very start, your experiment sounds interesting. Indeed, hypoxic training has been around for a while and subject to a lot of rumours, some unfounded, so studies like yours are useful. However, your experiment is marred by too many weaknesses. Here is a sample:
- Number of subjects (20 is way too few)
- For an experiment led on runners, exercising on an ergocycle is not representative of the constraints they face in competition. Hypoxic training on a treadmill would have been more apt
- ANS function values seem to decrease in subjects submitted to normoxic training, while it's clearly on the increase after hypoxic training. You should explain this discrepancy.
- Figure 1 is nowhere to be seen (but this may not be your fault)
- You spend way too much time explaining why your experiment is relevant instead of showing its usefulness. Maybe it's because your results are not very significant?
- Runners' weights between table 1 and table 2 are significantly different.
- And this point may seem accessory but it's not: your runners do not appear very fit. 20% of body fat for a competitive male runner is really a lot. I mean, I myself am a competitive runner, well into my thirties, I have 10% of body fat and a BMI of 20,8 (and I sometimes train in hypoxia BTW). Your experiment was not a fitness contest, I agree, but the physical characteristics of your subjects change the scope of your study. You should be more specific about this as well.
Your article could still be published but you should be much less assertive i the conclusion and, on the other hand, insist on what is really of interest to the reader: the impact of hypoxia on immune function. Effects of hypoxia on the heart or on performance are already well-documented and, even if I don't have access to fig 1, the physical profile of your subjects implies that they are not good enough runners for their athletic performances to carry real significance.
Eventually, there are few corrections to be made in the text itself:
- "Enhancing these factors related to endurance exercise performance and athletic performance by improving time until fatigue and increasing exercise intensity increase the efficiency of aerobic energy production and consequently enhance the maximum oxygen uptake (VO2max)" (lines 35 to 37). This sentence is much too complicated and I'm not even sure to understand its meaning.
- "a more improved tendency" (l.200-201): it's not idiomatic.
- Line 313: O2. "2" should be in lower case.
- And again, figure 1 got lost in the process...
I hope my advice will be useful.
Regards,
Author Response
Point-by-Point Responses to the Reviewers
We thank the reviewers for their guidance for further improving our revised manuscript (ijerph-731013) entitled “Interval hypoxic training enhances athletic performance and does not adversely affect immune function in middle- and long-distance runners”. As described below, we have responded to all the comments brought up by the reviewers and incorporated all the changes suggested by the reviewer. In addition, we have received English corrections from the English grammer editing company, Editage.
Reviewer 3.
Comments and Suggestions for Authors
Dear authors,
In this article, you gave me the impression of trying to reinvent the wheel. It's a pity because at the very start, your experiment sounds interesting. Indeed, hypoxic training has been around for a while and subject to a lot of rumours, some unfounded, so studies like yours are useful. However, your experiment is marred by too many weaknesses. Here is a sample:
Number of subjects (20 is way too few)
: We also agree with reviewer 3's comments. However, we validated the normality of the distribution of all outcome variables using the Kolmogorov-Smirnov test and calculated the sample size as follows: An a priori power analysis was performed with G-power for the energy metabolic parameter (VO2 during 30-min of submaximal exercise) based on previous research [1], indicating that a sample size of 14 participants (7 subjects per group) would be required to provide 80% power at an α-level of 0.05. We anticipated a more than 10% dropout rate and aimed for a starting population of 20. Nevertheless, we added sample size to the limitations of the study as follows: Although present studies have been designed systematically with equally controlled experiments, small sample sizes can be limited to check the effects of an intermittent interval training in a hypoxic condition vs a normoxic condition on athletic performance, hemodynamic function, ANS function, and immune function in middle- and long-distance competitive runners. More subjects may be needed in future studies to access sports field practice.
For an experiment led on runners, exercising on an ergocycle is not representative of the constraints they face in competition. Hypoxic training on a treadmill would have been more apt
: This is our big mistake. In our study, the interval training sessions consisted of 10 repetitions of interval running exercise (5 min of exercise corresponding to 90–95% HRmax and 1 min of rest) with treadmill. The ergocycle was only used for the measurement of hemodynamic function during submaximal exercise. It is a mistake in writing manuscript.
ANS function values seem to decrease in subjects submitted to normoxic training, while it's clearly on the increase after hypoxic training. You should explain this discrepancy.
: In general, exercise training in hypoxic and normoxic conditions has been reported to produce positive changes in HRV parameters at rest. However, improvements in HRV parameters are known to be more evident by hypoxic training. Although the subjects are very different, Park et al. (2019) first investigated the effect of hypoxic training on HRV and salivary cortisol in obese older men, and verified that exercise modality in hypoxia increases HF and decreases the LF/HF ratio and salivary cortisol compared with normoxic training. After post-hoc analysis, the improvement of HF and LF / HF ratio was shown only in hypoxic training group. However, our study found that the ANS function represented by HRV in the NTG decreased. This is in contrast to previous studies that exercise training improves ANS function. This decrease in ANS function cannot be interpreted accurately because the dietary intake and conditioning analysis of the athletes are not performed, but it is probably the result of poorly maintained conditioning during training. In the future, we believe that research should be conducted to accurately interpret this. As a results, the lack of accurate interpretation of ANS function reduction by normoxic training is a major limitation of this study.
Reference
Park HY, Jung WS, Kim J, Lim K. Twelve weeks of exercise modality in hypoxia enhances health-related function in obese older Korean men: A randomized controlled trial. Geriatr. Gerontol. Int. 2019;19:311–316. https://doi.org/10.1111/ggi.13625
Figure 1 is nowhere to be seen (but this may not be your fault)
: We added figure 1 to manuscript.
You spend way too much time explaining why your experiment is relevant instead of showing its usefulness. Maybe it's because your results are not very significant?
: There have been many researches on improving exercise performance by interval hypoxic training. The utility of this is now a universal result. However, few studies have examined the effectiveness of interval hypoxic training in terms of HRV parameters and immune function. So, we spent a lot of time explaining why our research designs are universal. We believe that this process is necessary for the average reader.
Runners' weights between table 1 and table 2 are significantly different.
: We are very sorry and this is our big mistake. There was a big mistake in our data processing. We reconfirmed the raw data and entered the correct results in all tables.
And this point may seem accessory but it's not: your runners do not appear very fit. 20% of body fat for a competitive male runner is really a lot. I mean, I myself am a competitive runner, well into my thirties, I have 10% of body fat and a BMI of 20,8 (and I sometimes train in hypoxia BTW). Your experiment was not a fitness contest, I agree, but the physical characteristics of your subjects change the scope of your study. You should be more specific about this as well.
: Thanks for the good information and comments. We also agree with reviewer 3's comments. As shown in the modified table 1, Runners in our study had a percent of body fat of 17-18% and a BMI of 22-23. These percent of body fat and BMI are relatively high on runners. However, as a result of examining the body composition of athletes belonging to the Korea Athletic Federation, excluding elite athletes, it was confirmed that the percentage of body fat was around 15%. The runners who participated in this study were athletes belonging to the Korean Federation of Athletic Federations, but they were relatively high in percent of body fat because they were in the off-season and moderately trained runners. We believe this is also a limitation of this study.
Your article could still be published but you should be much less assertive i the conclusion and, on the other hand, insist on what is really of interest to the reader: the impact of hypoxia on immune function. Effects of hypoxia on the heart or on performance are already well-documented and, even if I don't have access to fig 1, the physical profile of your subjects implies that they are not good enough runners for their athletic performances to carry real significance.
: We made the following changes to the conclusions; our study confirmed that intermittent interval training in a hypoxic condition for 6 weeks may effective in enhancing athletic performance and improving hemodynamic function and ANS function and does not adversely affect the immune function of competitive runners in comparison with training in a normoxic condition. We also insisted the following effects of the IHT on immune function; given that WADA is concerned that various hypoxic training regimes can have a potentially negative effect on health, our findings are of great value for athletes in performing hypoxic training. And once again, they were relatively high in percent of body fat because they were in the off-season and moderately trained runners.
Eventually, there are few corrections to be made in the text itself:
: We have received English corrections from the English grammer editing company, Editage.
"Enhancing these factors related to endurance exercise performance and athletic performance by improving time until fatigue and increasing exercise intensity increase the efficiency of aerobic energy production and consequently enhance the maximum oxygen uptake (VO2max)" (lines 35 to 37). This sentence is much too complicated and I'm not even sure to understand its meaning.
: Changed this sentence to the following; Enhancing these factors related to endurance exercise performance increases the efficiency of aerobic energy production and consequently enhances the maximum oxygen uptake (VO2max), as well as enhancing the athletic performance by improving time until fatigue and increasing exercise intensity [3,4].
"a more improved tendency" (l.200-201): it's not idiomatic.
: The sentence has been changed as follows; the HTG showed the potential to be improved did than the NTG (NTG: -4.4%, HTG: -5.4%).
Line 313: O2. "2" should be in lower case.
: As reviewer 3 comments, we corrected it.
And again, figure 1 got lost in the process...
: We added figure 1 to manuscript.
I hope my advice will be useful.
Round 2
Reviewer 3 Report
Dear authors,
My profound opinion on the study does not really change, however you have cleared the majority of my doubts.
I'm still not convinced this article is perfectly relevant, but you mention its limitations in the end and have shown honesty and goodwill, so according to me, it's fit for publication as long as you change the words in italics below:
l. 208: "the HTG showed the potential to be improved did than the NTG (NTG: -4.4%, HTG: -5.4%)". "To be improved did" does not mean anything.
l. 341: "Our study proven that intermittent interval training in a hypoxic condition more improves SDDN, RMSSD, HF band, and LF/HF band ratio compared with in a normoxic condition."
"Compared with in stg" in not correct.
Regards,
Author Response
I'm still not convinced this article is perfectly relevant, but you mention its limitations in the end and have shown honesty and goodwill, so according to me, it's fit for publication as long as you change the words in italics below:
1. 208: "the HTG showed the potential to be improved did than the NTG (NTG: -4.4%, HTG: -5.4%)". "To be improved did" does not mean anything.
-> Thank you very much for your comments. For elite athletes, it's very hard to bring about an improved in physical performance. Although there is no significant difference, I think that it is effective to show a tendency to decrease. Therefore, this study presented as follows; "No significant interaction was observed for the 3000-m time trial records; however, the HTG showed the potential to be effective did than the NTG (NTG: -4.4%, HTG: -5.4%)."
2. 341: "Our study proven that intermittent interval training in a hypoxic condition more improves SDDN, RMSSD, HF band, and LF/HF band ratio compared with in a normoxic condition."
"Compared with in stg" in not correct.
-> Thank you very much for your comments. As suggested by the reviewer has been deleted and modified as follows: "Our study proven that intermittent interval training in a hypoxic condition more improves SDDN, RMSSD, HF band, and LF/HF band ratio."